# $P^2SAM$: Probabilistically Prompted SAMs Are Efficient Segmentator for Ambiguous Medical Images

## ABSTRACT

The ability to generate an array of plausible outputs for a single input has profound implications for dealing with inherent ambiguity in visual scenarios. This is evident in scenarios where diverse semantic segmentation annotations for a single medical image are provided by various experts. Existing methods hinge on probabilistic modelling of representations to depict this ambiguity and rely on extensive multi-output annotated data to learn this probabilistic space. However, these methods often falter when only a limited amount of ambiguously labelled data is available, which is a common occurrence in real-world applications. To surmount these challenges, we propose a novel framework, termed as $P^2SAM$, that leverages the prior knowledge of the Segment Anything Model (SAM) during the segmentation of ambiguous objects. Specifically, we delve into an inherent drawback of SAM in deterministic segmentation, i.e., the sensitivity of output to prompts, and ingeniously transform this into an advantage for ambiguous segmentation tasks by introducing a prior probabilistic space for prompts. Experimental results demonstrate that our strategy significantly enhances the precision and diversity of medical segmentation through the utilization of a small number of ambiguously annotated samples by doctors. Rigorous benchmarking experiments against state-of-the-art methods indicate that our method achieves superior segmentation precision and diversified outputs with fewer training data (using simply 5.5% samples, +12% $D_{max}$). The $P^2SAM$ signifies a substantial step towards the practical deployment of probabilistic models in real-world scenarios with limited data.

## CCS CONCEPTS

• **Computing methodologies → Image segmentation**.

## KEYWORDS

Probabilistic modeling, Medical image segmentation, Prompting for foundation model

## 1 INTRODUCTION

Numerous complex situations in the physical domain often encompass a spectrum of potentially viable solutions. This is particularly noticeable in medical imaging, where inherent ambiguity in boundary structures and multiple plausible annotations arise due to limitations in imaging mechanisms, indeterminate boundaries among

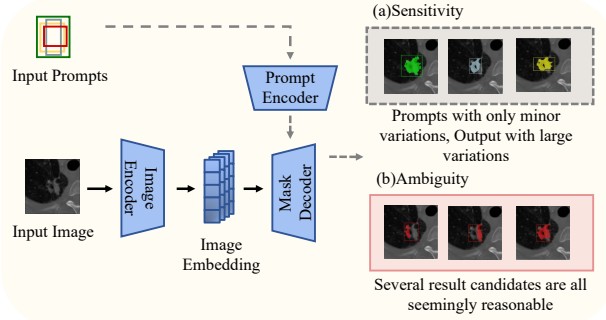

**Figure 1: The challenges exhibited by SAM in deterministic segmentation tasks. a) High sensitivity of SAM output to input prompts. SAM generates significantly diverse segmentation results while the prompt box is simply subtly varied. b) Ambiguity in SAM output for a given prompt. Multiple segmentation results all exhibit to be reasonable for a prompt especially when executing segmentation on an image with complex hierarchical structures.**

medical professionals, and varying experiences. The task paradigm associated with such ambiguity in the data itself, defined as "single input, multiple outputs", is referred to as Ambiguous Segmentation [15]. The advantages of this paradigm are evident. For instance, providing multiple regions of a lesion automatically can assist doctors in focusing on the areas of concern, rather than on ambiguous regions. However, traditional models, which establish a one-to-one mapping between inputs and outputs [20, 38], generating a unique segmentation map for each image, are fundamentally incapable of addressing such *ambiguous* scenarios.

To tackle the *ambiguity* issue, a variety of works have restructured the conventional "one-to-one" segmentation process by amalgamating insights from multiple experts and producing a range of outputs that account for pixel uncertainty and diverse image annotations. For instance, the Probabilistic U-Net [15], which combines U-net [33] and cVAE [13, 37], effectively encapsulates annotation distributions to generate an array of segmentation maps. Concurrently, models such as PHiSeg [2], PixelSeg [47], and CIMD [31] tackle uncertainty via varied sampling and introduce new accuracy metrics for segmentation.

Despite these advances, current methods still struggle to balance segmentation fidelity and diversity. This is primarily attributed to the fact that these probabilistic modeling techniques often sacrifice prediction accuracy to increase the complexity of distribution space and generate more diverse annotations. Additionally, compared to learning conventional deterministic mappings, probabilistic modeling inherently requires more training samples to fit an underlying "one-to-many" distribution of uncertainty. However, in actual

clinical diagnoses, there is often a shortage of high-quality lesion samples annotated by multiple experts, leading to suboptimal probabilistic modeling. To alleviate this degradation of performance in practical applications, this paper presents a pioneering study for probabilistic and ambiguous representation learning under data-limited settings.

Our insights are influenced by the remarkable achievements of Visual Foundation Models (VFM) [7, 30, 40], specifically SAM [14], which have been pre-trained on more than a billion masks across eleven million natural images. These models leverage prior knowledge extracted during large-scale pre-training to facilitate the segmentation of downstream tasks. Despite SAM's impressive generalization capabilities in segmentation, certain constraints have been observed: (1) SAM exhibits notable sensitivity to minor variations in prompts, necessitating precision in prompt inputs as illustrated in Figure 1(a), where minor translational or scaling operations on the input detection box prompt can cause significant alterations in SAM's output. (2) SAM may grapple with the issue of prompt ambiguity when confronted with target objects possessing complex hierarchical structures. This is due to the challenges in defining boundaries of elements at different levels, all of which seem feasible for a given prompt, as shown in Figure1(b). This dilemma requests SAM to generate multiple deterministic segmentation mask candidates.

In this work, we explore an unconventional perspective in an attempt to turn the *ambiguity* disadvantage of SAM in a deterministic segmentation task into an advantage in probabilistic ambiguous segmentation. Specially, we aspire to address the following two pertinent questions. First, considering the sensitivity of SAM's output to prompts, how can we model the distribution of prompts to control the generation of diversified segmentation outputs? Second, given the ambiguity of SAM's output to prompts, how can we modulate these ambiguous outputs as a reference for probabilistic modeling segmentation?

Inspired by these insights, we propose a Probabilistically Prompted Segment Anything framework, dubbed as $P^2SAM$, which adeptly addresses ambiguous segmentation tasks in medical imaging. Specifically, we initially design a hint generation network for probabilistic modeling of the representation of prompts input into SAM, simulating the modulating effect of different prompts on the output results. After sampling from the aforementioned prompt distribution, we employ a diversity-aware ambiguous ensemble algorithm to adaptively perceive the optimal ensemble weights of diverse ambiguous segmentation outputs, further modulating the multiple ambiguous segmentation masks generated by SAM. Lastly, by integrating the aforementioned strategies into the efficient fine-tuning of the current state-of-the-art medical SAM framework, our framework demonstrates powerful data efficiency in carrying out ambiguous segmentation tasks. In summary, our contributions can be outlined as follows:

- We introduce a Probabilistically Prompted Segment Anything Model ($P^2SAM$) that leverages not only the powerful segmentation prior knowledge inherent in SAM but also exploits the latent uncertainty knowledge yet to be discovered in SAM.

- We architect a generative network to probabilistically model the prior distribution of prompts, which conditions on the input image to generate meaningful distributions of prompts usable by SAM.
- We develop a diversity-aware ambiguous ensemble algorithm that effectively guides the model to adaptively perceive the contribution of SAM's different masks, thus significantly promoting the diversity of segmentation masks.
- Through extensive empirical benchmarking, our method significantly outperforms previous state-of-the-arts in terms of both accurate segmentation and diverse prediction that approaches the physicians, while using much less training samples at the same time.

## 2 RELATED WORK

**Ambiguous Image Segmentation.** Ambiguous image segmentation methodologies aim to encapsulate the aleatoric uncertainties and inherent unpredictability of labels employed for segmentation. A plethora of research has proposed diverse techniques to quantify aleatoric uncertainty. Preliminary research focused on enhancing a conventional U-Net[4, 10, 34, 43] with a probabilistic component to generate multiple predictions for an identical image, typically achieved by incorporating a conditional variational autoencoder (cVAE) [36]. The cVAE's low-dimensional latent space encodes potential segmentation variants. In [15], samples from this latent space are upscaled and concatenated at the U-Net's final layer. Numerous methodologies extend this setup to a hierarchical variant [2, 16, 48]. Other research utilises normalising flows to allow for a distribution more expressive than the Gaussian distribution in the cVAE [35, 39], switch to a discrete latent space [29], or incorporate variational dropout and directly use inter-grader variability as a training target [9]. Several other methods do not rely on the Probabilistic U-Net [5, 12, 21, 27, 44]. Monteiro *et al.* [27] propose a network utilising a low-rank multivariate normal distribution to model the logit distribution. Kassapis *et al.* [12] leverage adversarial training to learn potential label maps based on the logits of a trained segmentation network. Zhang *et al.* [47] employ an autoregressive PixelCNN to model the conditional distribution between pixels. Lastly, Gao *et al.* [6] use a mixture of stochastic experts, where each expert network estimates a mode of uncertainty, and a gating network predicts the probabilities that an input image is segmented by one of the experts. Different from previous efforts, our methodology is the inaugural exploration of employing large-scale pre-trained models for ambiguous image segmentation.

**Prompting Segmentation Foundation Models.** In recent years, the potential of large-scale vision models for image segmentation has been demonstrated by several concurrent works, inspired by language foundation models [3, 17, 24, 46]. These Segmentation Foundation Models (SFMs) like the Segment Anything Model (SAM) [14] and SEEM [50], have showcased impressive segmentation performance across diverse downstream datasets. SAM, utilizing a data engine with a model-in-the-loop annotation, learns a promptable segmentation framework that generalizes to downstream scenarios in a zero-shot manner. Other models like Painter [41] and SegGPT [42] introduce a robust in-context learning paradigm and can segment any images given an image-mask prompt. SEEM [50], on the

other hand, presents a general segmentation model prompted by multi-modal references, such as language and audio, incorporating versatile semantic knowledge. These advances in SFMs, largely driven by the *promptable segmentation* design, involve two types of prompts: semantic prompts (e.g., free-form texts) and spatial prompts (e.g., points or bounding boxes) [14, 25, 45, 50]. Depite these advances, acquiring suitable prompts for SFMs remains a largely under-explored area. This work aims to investigate the generation of effective prompts for SAM, with a focus on utilizing pre-training knowledge to complete ambiguous image segmentation.

## 3 METHOD

### 3.1 A Revisit of Segment Anything Model (SAM)

Segment Anything Model (SAM), an exemplar of transformer-based architecture, has demonstrated remarkable efficacy in the realms of natural language processing and image recognition tasks. Specifically, SAM employs a vision transformer-based image encoder to extract salient image features, prompt encoders to assimilate user interactions, and subsequently, a mask decoder to generate segmentation results and confidence scores, contingent on the image embedding, prompt embedding, and output token. SAM is a tripartite structure comprising of a prompt encoder, an image encoder, and a lightweight mask decoder, denoted respectively as $\text{Enc}_P$, $\text{Enc}_I$, and $\text{Dec}_M$. As an interactive framework, SAM ingests an image $I$, and a set of prompts $P$, which may be a point, a box, or a coarse mask. Specifically, SAM first employs $\text{Enc}_I$ to obtain the input image feature, and adopts $\text{Enc}_P$ to encode the human-given prompts of a length $k$ into prompt tokens as follows

$$F_I = \text{Enc}_I(I), \quad T_P = \text{Enc}_P(P), \tag{1}$$

where $F_I \in \mathbb{R}^{h \times w \times c}$ and $T_P \in \mathbb{R}^{k \times c}$, where the resolution of the image feature map is represented by $h$, $w$, and the feature dimension is denoted by $c$. Subsequently, the encoded image and prompts are introduced into the decoder $\text{Dec}_M$ for interaction based on attention mechanisms. SAM constructs the decoder's input tokens by concatenating several learnable mask tokens $T_M$ as prefixes to the prompt tokens $T_P$. These mask tokens are accountable for generating the mask output, formulated as follows

$$M = \text{Dec}_M\left(F_I, \ \text{Concat}(T_M, T_P)\right), \tag{2}$$

where $M$ denotes the final segmentation mask predicted by SAM.

### 3.2 Lifting SAM to Probabilistic Space

Ambiguous segmentation tasks require multiple segmentation results for a single input to more accurately reflect the true distribution of real-world scenarios. Interestingly, we observe an inherent ambiguity in SAM, where minor positional modifications to prompts lead to substantial alterations in SAM's segmentation output. This observation catalyzes our consideration for probabilistic modeling of prompt variations. By utilizing a distribution of prompt embedding, rather than a single deterministic prompt, we can effectively modulate the model output, as

$$\tilde{T}_P \sim \mathcal{P}_{PE}(\theta), \tag{3}$$

where $\mathcal{P}_{PE}$ denotes a probability distribution for the space of prompt embedding, $\tilde{T}_P$ is specific a prompt sampling from the given distribution at one time. Formally, by implementing multiple rounds of sampling, we can construct a probabilistic mapping of segmentation outputs with respect to their prompts, formulated as the format of expectation

$$\mathbb{E}_{\tilde{M} \sim \mathcal{P}_M(\vartheta)} = \mathbb{E}_{\tilde{T}_P \sim \mathcal{P}_{PE}(\theta)} \text{Dec}_M\left(F_I, \ \text{Concat}(T_M, \tilde{T}_P)\right) \tag{4}$$

where $\tilde{M}$ denotes the SAM output corresponding to the prompt sampling every time, which can also be interpreted as the sampling from a virtual distribution $\mathcal{P}_M$ for the segmentation results which obeys the parameters $\vartheta$. As a result, we can construct an optimized probability distribution $\tilde{T}_P \sim \mathcal{P}_{PE}(\theta)$ by narrowing the gap between $\tilde{M} \sim \mathcal{P}_M(\vartheta)$ and the ground-truth distribution.

### 3.3 Instance-conditional Probabilistic Prompt Generation

To model the probability distribution of prompt embedding, it is imperative to estimate the parameters $\theta$ of this distribution. We adopt an axisymmetric Gaussian distribution to characterize the prompt embedding, which is dictated by two crucial parameters: $\mu$ (mean) and $\sigma$ (standard deviation). To accurately model the prompt embedding, we have designed a dedicated prompt generation network. This network comprises two primary components: an encoder and an Axis-Gaussian generation network. The encoder, composed of several simple convolution blocks, is designed to extract image features. Subsequently, these feature maps are introduced to the Axis-Gaussian generation network, which is a convolutional network structure [49]. Then we can sample a prompt embedding from the given Gaussian distribution by

$$\tilde{T}_P \sim \mathcal{N}(\mu_I, \text{diag}(\sigma_I)), \tag{5}$$

where $\mu_I$ and $\sigma_I$ denotes the parameters characterized for image $I$.

We further dedicated a Prompt Generation Network, made up of several straightforward convolutional blocks, aims to extract features from the image. Subsequently, these feature maps are fed into the Axis-Gaussian generation network, which is also a convolutional network structure [49]. Considered the variation in salient regions within an image suggests that the required prompt location and size should also differ, making it impractical to apply a uniform probability distribution model to prompt embeddings. Hence, we introduce image prior knowledge into the Prompt Generation Network during forward inference. By incorporating this prior knowledge, the network can customize a unique axis Gaussian distribution for each image $I$, thus achieving more precise sampling for the prompt embedding, as

$$\mu_I, \sigma_I = PGN(\theta|I), \tag{6}$$

where PGN stands for the Prompt Generation Network modeled by the parameters $\omega$, and $\mu$ and $\sigma$ respectively denote the mean and standard deviation of the Axis Gaussian Distribution generated by the network, where $\mu, \sigma \in \mathbb{R}^N$ with N=256.

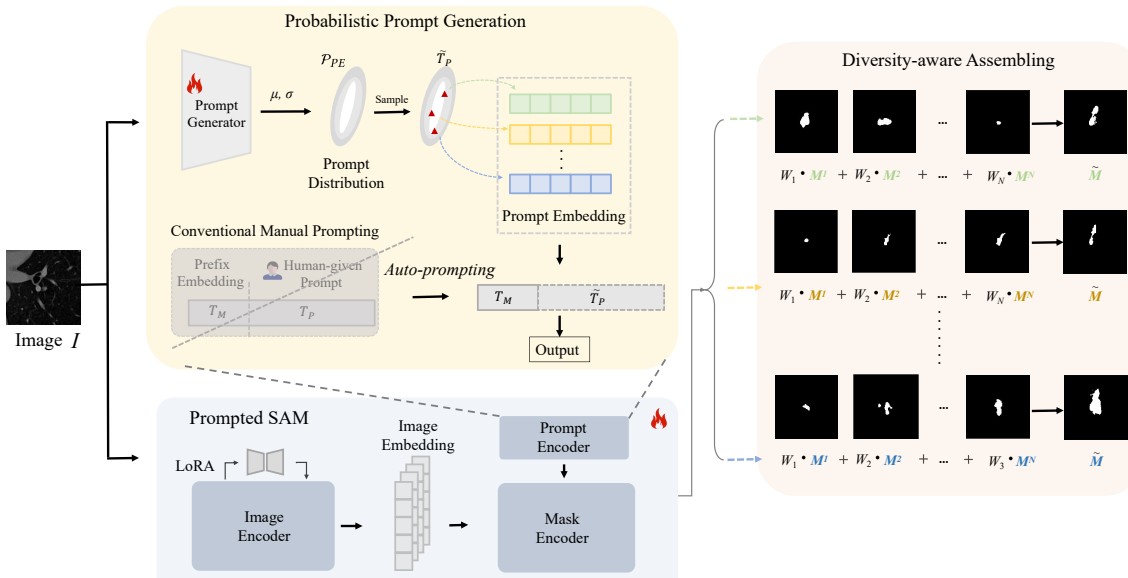

**Figure 2: P$^2$SAM Training Pipeline. We first lift the conventional SAM prompting to the probabilistic space, by leveraging a network targeting at generating prompt distribution. Then we sample the prompt embedding from the probabilistic latent space and instill it into SAM to unlock the capacity of SAM in "one-to-many" ambiguous segmentation. We carefully design a diversity-aware assembling that perceives the inherent diversity in SAM and turn it to ensembled ambiguous output.**

## 3.4 Diversity-aware Assembling

We assume that SAM implicitly models the probability of default adaptive prompts in Section 3.2, but how does this adaptive prompt focus on images with complex hierarchical structures? When dealing with images with complex hierarchical structures, it is not clear how this adaptive prompt effectively focuses on key areas. Especially when SAM faces segmentation tasks on an image containing multiple salient targets, it often faces the challenge of segmentation ambiguity. To overcome this ambiguity, SAM generates multiple segmentation masks to segment the salient regions of the image from different levels and perspectives. Although this method can provide multi angle segmentation results, these results often fail to fully reflect the certainty and uniqueness of segmentation, making it difficult to provide a more convincing and clear segmentation.

To integrate the multi-scale segmentation masks of SAM under ambiguous prompts, we introduce a ambiguous integration strategy with diverse sensitivities. This strategy relies on SAM to obtain multi-scale outputs, which refer to the original segmentation results of multi-scale output by SAM, as $\{M^1, M^2, ..., M^N\}$, where $N$ denotes the number of scale. On top of this, we adopt learnable mask weights $\mathcal{W} = \{w_1, w_2, ..., w_N\} \in \mathbb{R}^N$, and calculate final mask output through weighted summation as:

$$\tilde{M} = \Sigma_{i=1}^N w_i * \tilde{M}^i \qquad (7)$$

In order to learn the optimal weights, we fine-tune SAM and also trained this parameter. By adopting this strategy, we can effectively learn and understand the scale perception of objects while preserving the deep knowledge of the pre-trained model. In addition, it can adaptively integrate masks of multiple scales to achieve precise output of the optimal segmentation scale for the target object.

## 3.5 Overall Optimization Procedure

During the optimization of the entire framework, we found that the direct application of SAM is limited in certain specific vertical scenarios. Therefore, we propose to fine-tune SAM to our tasks first, followed by efficient probabilistic prompt training. Specifically, the overall optimization process is divided into two crucial stages. In the first stage, we aim to fine-tune the key modules within the SAM model to empower its adaptation ability [19, 22], including the modulation module, which integrates diverse outputs, the mask decoder, the prompt encoder, and the image encoder. Notably, we fine-tune the image encoder via the Low Rank Adaptation (LoRA) [8] strategy, thereby keeping the original parameters of the image encoder unchanged. Specially, at the data level, we only use non-empty labels for model fine-tuning and training, which speeds up the model's adaptation. The loss function used in this process is as follows:

$$\mathcal{L}_1(\tilde{M}, \tilde{GT}; Enc_P, Dec_M, \mathcal{W}, Enc_I(LoRA)) = \ell_{seg}(\tilde{M}, \tilde{GT}), \quad (8)$$

At the second stage, upon establishing the benchmark performance for the segmentation task and the capability to handle ambiguous sets, we further enhanced our model to address the challenges associated with ambiguous segmentation. In this stage, we froze the parameters for all components and concentrated on training the prompt generation network. The image is provided as input

to the prompt generation network, which is responsible for generating a precise prompt probability distribution. Following this, a series of approximate yet distinct prompts are obtained by sampling from this distribution. These prompts are then fed into the SAM model to achieve ambiguous segmentation. During this process, we employed the following loss function:

$$\mathcal{L}_2(\tilde{M}, \tilde{GT}; PGN) = \ell_{seg}(\tilde{M}, \tilde{GT}). \quad (9)$$

## 4 EXPERIMENT

### 4.1 Datasets

**Lung lesion segmentation (LIDC-IDRI).** This dataset is publicly accessible and comprises a substantial collection of 1018 lung Computed Tomography (CT) scans, derived from 1010 distinct subjects. This dataset is notable for its inclusion of manual annotations, contributed by a panel of four domain experts. This feature makes the dataset a robust and accurate reflection of the typical ambiguity often encountered in CT imaging, as referenced in the study [1]. A diverse group of 12 radiologists lent their expertise to provide annotation masks for this dataset, further enhancing its value. The version of the dataset we use in this study is the one obtained after the second reading. In this phase, the domain experts were presented with the annotations made by other radiologists. This process allowed them to make new adjustments based on the feedback and insights of their peers, thereby ensuring the dataset's annotations are comprehensive, accurate, and reflective of a broad spectrum of expert opinions.

**Brain tumour segmentation in 3D (BraTS 2017).** The BraTS 2017 [11] dataset encompasses 285 cases of 3D MRI images, each comprising 155 slices. Every slice is provided in four modalities (T1, T1ce, T2, and Flair) and has been meticulously annotated across four classes by expert radiologists: background (BG), non-enhanced/necrotic tumor core (NET), oedema (OD), and enhanced tumor core (ET). We overlay and amalgamate annotations from these various categories, transforming the results into a binary mask that solely includes the foreground and background. This procedure is designed to generate multiple segmentation masks to mimic actual ambiguous segmentation scenarios, thereby enhancing the rigor and reliability of the experiment.

### 4.2 Implementation Details

In our experiments, we utilized the LIDC dataset, provided by Probability U-net [15], which we partitioned into training, validation, and testing sets in a 60:20:20 ratio. During the initial stage of our experiment, we fine-tune the SAM using non-empty samples from the dataset, enabling us to learn weight modules for training. To accommodate the segmentation regions within the image, we configured SAM for multi-output mode and set the output quantity to 8, simultaneously employing default prompts to enhance the model's adaptive capture capability of the image. We initialized 8 learnable weights in the diversity-aware assembling module, each weighing 1/8, and supervised learning was performed on the final output synthesized by weighting these weights with a randomly selected non-empty label. At this juncture, we employed the Adam optimizer for optimization with a learning rate of 1e-3, and executed 100 epochs on the dataset. In the second stage, we froze the

parameters of SAM and the learnable weight modules, and focused solely on training the prompt generator network. At this point, all labels, including empty ones, were used for training in order to enable the model to more accurately establish the probability distribution of prompts. To manage the complexity of the prompt generator network and optimize its performance, we introduced a *L*2 regularization term to the loss function. During this stage, we continued to use the Adam optimizer, but adjusted the learning rate to 1e-5. Unless otherwise stipulated, we will report the performance results on the test set based on the model with the minimum loss on the validation set.

When comparing the BraTS 2017 dataset with the LIDC dataset, we noticed that the BraTS 2017 dataset provides higher image resolution (240x240 pixels). In order to adapt to the input specifications of the model, we uniformly adjusted all images to a size of 128x128 pixels, and we only used T1 mode for experiments. In addition, the dataset is also divided into training, validation, and testing sets in a ratio of 60:20:20. However, during the experiment, we only selected the first 500 samples from the training and testing sets for model training and evaluation [18, 23, 28]. Specifically, considering that each lesion tissue has three non overlapping annotations, in order to simulate the effect of ambiguous segmentation, we stacked these three sets of annotations in sequence, ultimately generating three layers of gradually expanding and overlapping lesion annotations. In order to optimize data quality, we removed samples containing empty annotations and slices, and ultimately obtained 8270 high-quality slice samples, each with three annotations attached. The training process is also divided into two stages, and since there is no empty annotation problem in this dataset, all annotation information is used throughout the entire training process. During the training process, we use the Adam optimizer to set the learning rate to 1e-3 in the first stage and adjust it to 1e-5 in the second stage.

### 4.3 Evaluation Metrics

**Generalized Energy Distance (GED).** A commonly used metric in ambiguous image segmentation tasks that leverages distance between observations by comparing the distribution of segmentations [15], as

$$D_{GED}^2(P_{gt}, P_{out}) = 2\mathbb{E}[d(S, Y)] - \mathbb{E}[d(S, S')] - \mathbb{E}[d(Y, Y')], \quad (10)$$

where, $d$ corresponds to the distance measure $d(x, y) = 1 - IoU(x, y)$, $Y$ and $Y'$ are independent samples of $P_{gt}$ and $S$ and $S'$ are sampled from $P_{out}$. Lower energy indicates better agreement between prediction and the ground truth distribution of segmentations.

**Maximum Dice Matching ($D_{max}$).** In medical diagnosis cases, empty sets, which indicate no abnormalities are also valid diagnoses. However, in this case, the Dice metric will be undefined, hence we set Dice = 1 in those cases. Specially, the Dice score is defined as:

$$Dice(\hat{Y}, Y) = \begin{cases} \frac{2|Y \cap \hat{Y}|}{|Y| + |\hat{Y}|}, & \text{if } Y \cup \hat{Y} \neq \emptyset \\ 1, & \text{otherwise.} \end{cases} \quad (11)$$

To calculate the best prediction accuracy for a set of prediction samples, we calculated the Dice score between each prediction result and each ground truth. We define the set of all Dice scores $\mathbf{D}_i$ for each individual ground truth $Y_i$, as follows

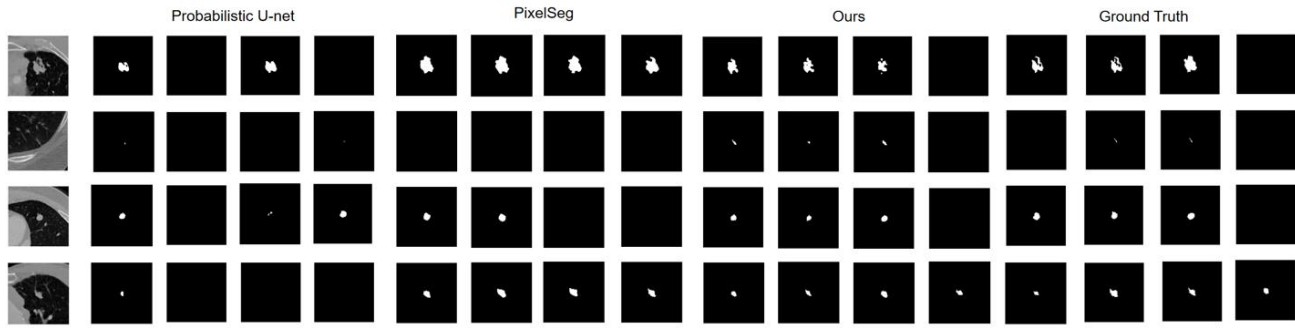

**Figure 3: Comparative qualitative analysis with the advanced methods, including Probabilistic U-net [15] and PixelSeg [47]. Examples of available four ground-truth expert labels and sampled segmentation masks are provided.**

**Table 1: Comparison of GED, HM-IoU, and $D_{max}$ quantitative results of LIDC(5.5% of the entire dataset) using state-of-the-art ambiguous segmentation networks.**

| Method | LIDC (500 samples) | | | |
|---|---|---|---|---|
| | GED16($\downarrow$) | GED32($\downarrow$) | HM-IoU($\uparrow$) | $D_{max}(\uparrow$) |
| Probabilistic U-net [15] | 0.325 | 0.337 | 0.324 | 0.251 |
| CAR [12] | 0.8849 | 0.905 | 0.179 | 0.567 |
| PixelSeg [47] | 0.328 | 0.299 | 0.495 | 0.731 |
| Mose [6] | 0.290 | 0.276 | 0.510 | 0.652 |
| P$^2$SAM (Ours) | **0.208** | **0.206** | **0.627** | **0.919** |

$$D_{max} = max\{Dice(\hat{Y}_1, Y_i), Dice(\hat{Y}_2, Y_i), ..., Dice(\hat{Y}_N, Y_i)\}, \quad (12)$$

where $\mathbf{D}_i$ is a collection of Dice scores calculated between each ground truth $Y_i$ and all the provided predictions. Then, we take the maximum dice score from this group and match it as the maximum dice score $D_{max}$.

**Hungarian-Matched Intersection over Union (HM-IoU).** GED excessively rewards sample diversity, which cannot reflect the sufficiency of the sample. Therefore, the Hungarian Matching IoU (HM-IoU) is proposed to calculate the optimal 1:1 between annotation and prediction, which better represents the fidelity of the sample. The Hungarian algorithm finds the optimal 1:1 match between objects in two sets, for which we use $IoU(Y, Y')$ to determine the similarity between the two samples.

### 4.4 Results on LIDC-IDRI

We evaluated our model on the LIDC-IDRI dataset which contains 1018 lung CT scans from 1010 patients and independently annotated by 12 radiologists. Each CT scan was annotated by 4 radiologists for multiple lung nodules. We use the pre-processed version provided by Probabilistic U-net [15], which extracted 15,096 slices of size 128×128 centred on the lesion and with four annotations per slice. Table 1 and Table 2 present the results. We used two data versions to train the model, which are 500 samples from the training set and all available training set samples.

We compare our approach to eleven recent stochastic segmentation methods: Probabilistic U-Net [15], Hierarchical Probabilistic U-Net (HProb. U-net) [16], PhiSeg [2], Stochastic Segmentation Network (SSN) [26], Calibrated Adversarial Refinement (CAR) [12], PixelSeg [47], Mixture of Stochastic Experts (MoSE) [6], and Collectively Intelligent Medical Diffusion (CIMD) [32].

We conducted performance evaluations on different ambiguous medical image segmentation models, taking into account multiple indicators such as GED, HM-IoU, and $D_{max}$. We annotate the metrics calculated with n samples using a subscript, *i.e.*, $GED_n$ and $HM-IoU_n$, and we set n to common values found in the literature. The above-mentioned metrics to measure the difference between the distributions of generated and ground truth label maps.

The results show that our method significantly outperforms other state-of-the-art networks in various metrics on two different training sample datasets. Specifically, a higher $D_{max}$ score indicates a high match between the distribution of the generated samples and the actual situation. Meanwhile, higher HM-IoU and lower GED scores comprehensively reflect the diversity and consistency of the samples, effectively quantifying the degree of agreement between prediction and annotation.

As the evaluation of ambiguous networks is difficult to characterize, we argue that qualitative results can be a good indicator of network performance, especially for difficult cases. We show the predictions from the test dataset for all the models in Figure 3. It can be seen that P$^2$SAM achieves visually superior and diverse

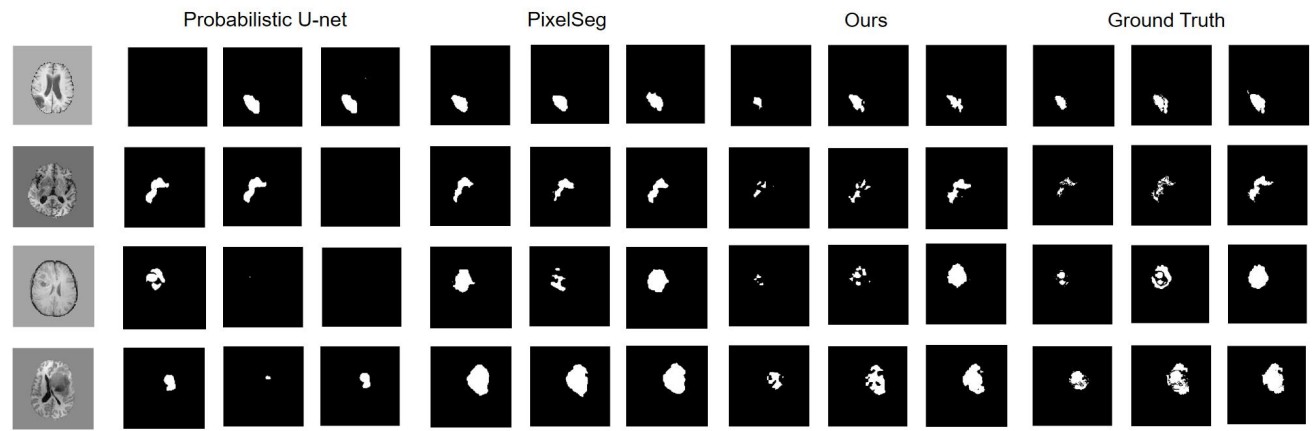

**Figure 4: Comparative qualitative analysis with the advanced methods including Probabilistic U-net [15] and PixelSeg [47]. Examples of the available three ground-truth expert labels and sampled segmentation masks are provided.**

**Table 2: Comparison of GED, HM-IoU, and $D_{max}$ quantitative results of LIDC (all samples) using state-of-the-art ambiguous segmentation networks.**

| Method | LIDC (all samples) | | | |
|---|---|---|---|---|
| | GED16($\downarrow$) | GED32($\downarrow$) | HM-IoU16($\uparrow$) | $D_{max}$($\uparrow$) |
| Probabilistic U-net [15] | 0.324 | 0.303 | 0.423 | 0.370 |
| HProb. U-net [16] | 0.270 | — | 0.530 | — |
| PHiseg [2] | 0.262 | 0.247 | 0.595 | — |
| SSN [26] | 0.259 | 0.243 | 0.555 | — |
| CAR [12] | 0.252 | — | 0.549 | 0.732 |
| PixelSeg [47] | 0.243 | 0.245 | 0.614 | 0.814 |
| CIMD [32] | 0.234 | 0.218 | 0.587 | — |
| Mose [6] | 0.234 | 0.230 | 0.623 | 0.702 |
| $P^2$SAM (Ours) | **0.218** | **0.216** | **0.679** | **0.933** |

**Table 3: Comparison of GED, HM-IoU, $D_{max}$ and $D_{mean}$ quantitative results of BraTS2017 using state-of-the-art ambiguous segmentation networks.**

| Method | BraTS2017 (500 samples) | | | | BraTS2017 (all samples) | | | |
|---|---|---|---|---|---|---|---|---|
| | GED($\downarrow$) | HM-IoU($\uparrow$) | $D_{max}$($\uparrow$) | $D_{mean}$($\uparrow$) | GED($\downarrow$) | HM-IoU($\uparrow$) | $D_{max}$($\uparrow$) | $D_{mean}$($\uparrow$) |
| Probabilistic U-net [15] | 0.154 | 0.427 | 0.517 | 0.346 | **0.225** | 0.521 | 0.645 | 0.464 |
| PixelSeg [47] | 0.549 | 0.414 | 0.516 | **0.373** | 0.419 | 0.528 | 0.785 | **0.561** |
| $P^2$SAM (Ours) | **0.134** | **0.435** | **0.730** | 0.363 | 0.238 | **0.593** | **0.881** | 0.494 |

results compared to the previous state-of-the-art methods. $P^2$SAM works especially well on ultrasound modalities with minimal error as can be seen in Figure 3. From Figure 3 it can be seen that $P^2$SAM is able to capture all the lesions even if they have small structures while maintaining diversity in segmentation masks. As $P^2$SAM injects stochasticity at each hierarchical feature representation, it demonstrates diverse and accurate segmentation in all datasets.

### 4.5 Results on BraTS2017

As demonstrated in Table 3, the quantitative outcomes of the $P^2$SAM, PixelSeg [47], and Probabilistic U-net [15] methodologies, when

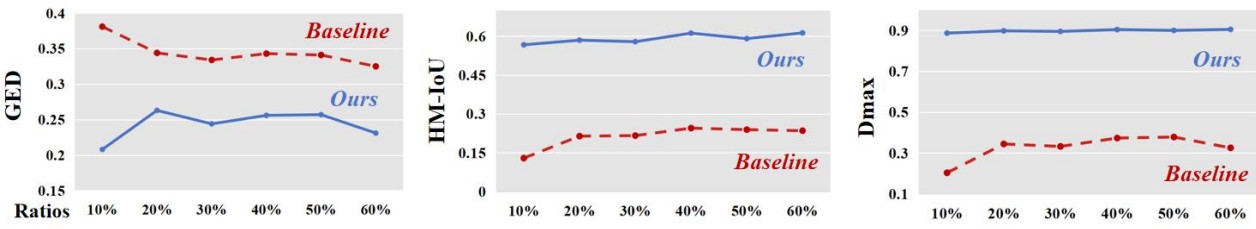

Figure 5: Comparison of GED ($\downarrow$), HM-IoU ($\uparrow$) and $D_{max}(\uparrow$) between ours and baseline models under five different data ratios on LIDC-IDRI dataset.

Table 4: Ablation study of the key strategies of the proposed P$^2$SAM on LIDC-IDRI dataset.

| Method | GED($\downarrow$) | $D_{max}(\uparrow$) | HM-IoU ($\uparrow$) |
|---|---|---|---|
| Vanilla Adapted SAM | 0.381 | 0.705 | 0.359 |
| SAM + Probabilistic Prompt | 0.340 | 0.803 | 0.454 |
| SAM + Diversity Assembling | 0.376 | 0.853 | 0.402 |
| P$^2$SAM (Full Model) | **0.208** | **0.919** | **0.627** |

employed on the BraTS 2017 dataset, exhibit significant differences. The P$^2$SAM methodology, in particular, stands out for its performance advantages in several key metrics, most notably the GED, $D_{max}$, and HM-IoU metrics. This underscores the effectiveness of the P$^2$SAM approach and its robustness when compared to the two baseline methods.

When considering the $D_{mean}$ metric, the P$^2$SAM approach performs on par with the two baselines, indicating a similar level of performance between the methodologies. This is a noteworthy observation as it suggests that even when trained with a limited dataset, P$^2$SAM is capable of generating a wider array of more accurate segmentation samples. This not only validates the efficiency of the P$^2$SAM methodology but also underscores its practical value in tackling tasks related to ambiguous medical image segmentation.

Moreover, the qualitative results, as depicted in Figure 4, offer further insights into the performance of the proposed method compared to other techniques. It becomes evident that the proposed method not only delivers more accurate and plausible segmentation results, but it also excels in generating a more diverse range of prediction results. This diversity in prediction outcomes is particularly valuable as it allows for a more comprehensive understanding and interpretation of the data, thereby enhancing the overall effectiveness of the segmentation process.

## 4.6 Ablation Study

In this section of ablation study, we deconstruct P$^2$SAM into three main components for in-depth analysis, including a fine tuned SAM model, learnable weights, and Prompt Generate Network (PGN). And we investigate the ability of each component to control the segmentation diversity and accuracy, by using the fine tuned SAM as a baseline. Scores are based on LIDC dataset.

*Vanilla Adapted SAM.* We first tested the fine tuned SAM model as a baseline, focusing on three key indicators: GED, $D_{max}$, and HM-IoU. The test results are detailed in Table 4.

*SAM + Probabilistic Prompt.* We introduced a prompt generator network on the fine tuned SAM to guide the generation of ambiguous prompts. Compared with the benchmark fine tuned SAM model, the introduction of PGN resulted in significant improvements in the three key performance indicators of GED, $D_{max}$, and HM-IoU, especially in the significant growth of GED. This result clearly indicates that the fusion of PGN not only enriches the output of the SAM model, but also significantly enhances the diversity of the output, further improving the performance of the model in ambiguous medical image segmentation tasks.

*SAM + Diversity Assembling.* We introduce learnable weights for diversity-aware assembling in our baseline model to guide the SAM model in outputting ambiguous segmentation results. Compared to the baseline model, there was little change in the GED indicator, but we observed significant improvement in the $D_{max}$ and HM-IoU indicators. This result clearly indicates that by using learnable weights modules, we can effectively integrate the multiple outputs of the SAM model, thereby generating samples that are both representative and more accurate.

*Full Model.* The full model (P$^2$SAM) achieves the best results when all components work together.

It appears that when any component is removed, the performance drops accordingly, revealing the effectiveness of our design.

## 4.7 Conclusion

This paper presents a novel framework, $P^2SAM$, to tackle the inherent ambiguity prevalent in real-world visual scenarios, particularly in medical image segmentation. By leveraging the prior knowledge of the Segment Anything Model (SAM) and transforming its inherent drawback into an advantage, we demonstrate significant improvements in the precision and diversity of medical segmentation. Despite the challenges posed by limited availability of ambiguously annotated samples, our method outperforms state-of-the-art methods in rigorous benchmarking experiments, achieving superior segmentation precision and diversified outputs with fewer training data. This signifies a substantial step towards the practical deployment of probabilistic models in real-world scenarios with limited data. Future work could be considered to further improving the performance of the probabilistic modeling and expanding its application to other tasks that requires to output ambiguous results.

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
