# OpenReview forum: "P^2SAM: Probabilistically Prompted SAMs Are Efficient Segmentator for Ambiguous Medical Images"
_acmmm.org/ACMMM/2024/Conference — MM2024 Poster_

### Official Review · Reviewer_xKcV · 2024-05-02

**Rating:** 5
**Confidence:** 3

**Summary:**

The article introduces a novel framework called P^2SAM to address inherent ambiguity in visual scenarios, particularly in medical image segmentation. It highlights the sensitivity and ambiguity disadvantage of SAM. P^2SAM leverages prior knowledge from the Segment Anything Model (SAM) to improve segmentation of ambiguous objects.

**Strengths:**

1. It is ineresting to highlight the two instinct characteristics of SAM, i.e., sensitivity and ambiguity, and transform it to the scenarios of ambiguous objects segmentation. The framework explores an unconventional perspective by leveraging the ambiguity disadvantage of SAM in deterministic segmentation tasks and turning it into an advantage in probabilistic ambiguous segmentation.
2. The writing is clear and concise, making it easy for readers to understand and follow the author's arguments.

**Limitations:**

1. It's better to introduce the background and definition of Ambiguous Medical Images more clearly.
2. Why set the output quantity to 8? Is this value has much influence on the performance?

**Suitability:**

2

---

### Official Review · Reviewer_W9qM · 2024-05-23

**Rating:** 4
**Confidence:** 3

**Summary:**

The paper introduces a novel framework, P2SAM, to address the inherent ambiguity in medical image segmentation. Leveraging the Segment Anything Model (SAM), P2SAM turns the sensitivity of SAM to input prompts from a disadvantage into an advantage for probabilistic modeling. By generating a distribution of prompts and employing a diversity-aware ambiguous ensemble algorithm, the framework enhances segmentation precision and diversity with limited training data. Extensive experiments demonstrate P2SAM's superior performance over state-of-the-art methods, especially in scenarios with minimal ambiguously annotated data​​.

**Strengths:**

1. Innovative Approach: The framework ingeniously transforms SAM's sensitivity to prompts into an advantage, allowing for effective ambiguous segmentation​​.
2. Probabilistic Modeling: P2SAM introduces a generative network for probabilistic prompt modeling, enhancing the ability to handle ambiguity​​.
3. Diversity-Aware Ensemble: The use of a diversity-aware ensemble algorithm improves the integration of multiple ambiguous outputs, leading to more accurate and diverse segmentation results​​.
4. Data Efficiency: The framework achieves superior performance with significantly fewer training samples compared to previous methods​​.
5. Empirical Validation: Extensive benchmarking against state-of-the-art methods confirms the effectiveness and robustness of P2SAM across different datasets​​.

**Limitations:**

I've noticed that recently there have been a lot of Super-Resolution methods to solve the problem of blurred or low-resolution images, and doing it directly to blurred images is also a good direction. I'm open-minded about this kind of work. Actually, i have some questions:
1. About Prompt Sensitivity: Although P2SAM leverages SAM's sensitivity to prompts for probabilistic modeling, this reliance may pose challenges when accurately defining prompts is difficult, especially in cases where prompt variability does not align well with the segmentation task​​.
2. About Limited Generalizability: Will there be domain gaps in ambiguous images from different machines?
3. About Data: How do you define blurry images?
4. About the Experiment: There are a few comparison methods in Table 3. Can we do more experiments for comparison?
5. Some measures in Table 3 do not reach SOTA. Can you explain why?

**Suitability:**

2

---

### Official Review · Reviewer_Rw9R · 2024-05-24

**Rating:** 2
**Confidence:** 3

**Summary:**

This paper introduces the Segment Anything Model (SAM) to perform segmentation for ambiguous medical images. It lifts the prompt to a prompt probabilistic distribution, which help generate multiple outputs.

**Strengths:**

The related work is comprehensive and the experimental results can achieve SOTA performance.

**Limitations:**

SAM is a interactive model and needs prompts to obtain the final outputs. This is relatively limited for medical image application, because the prompts commonly require professional medical knowledge which is difficult for ordinary people.

The paper does not interpret what kinds of prompts are applied in the proposed model (text, point or bbox). How do different kinds of prompts affect the final segmentation performance. Moreover, if the best results are achieved under points or bbox prompts, this is unfair for other comparison models. Actually, I think prior knowledge prompts (points or bbox) should not appear in the inference stage.

**Suitability:**

2

---

### Official Review · Reviewer_6Ggs · 2024-05-24

**Rating:** 3
**Confidence:** 3

**Summary:**

This paper introduces a framework called P² SAM, which utilizes prior knowledge from foundation models to segment ambiguous objects. Experimental findings indicate that employing a limited number of doctor-annotated ambiguous samples substantially improves both precision and diversity in medical segmentation.

**Strengths:**

The experimental outcomes demonstrate state-of-the-art (SOTA) performance.

**Limitations:**

The reported results for CIMD are not consistent with the original paper. This makes the experimental results not very convincing.

More datasets like Bone surface segmentation or Multiple sclerosis lesion segmentation can be applied to further demonstrate the effectiveness of the proposed model.

**Suitability:**

2

---

### Meta-Review · Area_Chair_sfk2 · 2024-07-09

**Recommendation:** Accept (Poster)
**Confidence:** 4

**Metareview:**

The reviewers acknowledge the contributions of this paper with the innovative transformation of SAM's characteristics into advantages for probabilistic modelling in ambiguous medical image segmentation. The final version should add more experimental comparisons, including SAMed.